# Recovery of K$_2$SO$_4$ and Separation of SiO$_2$/Al$_2$O$_3$ from Brown Corundum Fly Ash

**Yuandong Xiong** [1,2]**, Junqi Li** [1,2,*]**, Qian Long** [1,2]**, Chaoyi Chen** [1,2,*]**,
Yuanpei Lan** [1,2] **and Linzhu Wang** [1,2]

[1] College of Materials and Metallurgy, Guizhou University, Guiyang 550025, China;
ydxiong1996@126.com (Y.X.); long-qian@csu.edu.cn (Q.L.); yplan@gzu.edu.cn (Y.L.);
lzwang@gzu.edu.cn (L.W.)

[2] Guizhou Province Key Laboratory of Metallurgical Engineering and Process Energy Saving,
Guiyang 550025, China

\* Correspondence: jqli@gzu.edu.cn (J.L.); czchen@gzu.edu.cn (C.C.); Tel.: +86-1850-6801-5817 (C.C.)

**Abstract:** Brown corundum fly ash (BCFA), which is the collected ash in brown corundum production, has received lots of environmental concerns due to its ultra-fine particle size and complex composition. Aiming to recycle the major elements including K, Al and Si from BCFA environmentally, this work proposed a simple and non-pollution process to utilize BCFA by water leaching, size screening and solution evaporation. The influences of water leaching conditions including leaching temperature, time and liquid-to-solid ratio was considered to optimize the K$_2$SO$_4$ recovery efficiency. Results show that the potassium sulfate content of the product is 75.7% after water leaching at 60 °C for 15 min with the liquid–solid ratio of 20:1. The wet screening and size separation with a 1 μm sieve can separate and enrich aluminum and silicon significantly. The aluminum-rich product is composed of 54.65% of alumina and 11.04% silica, with the alumina and silica ratio (A/S) of 4.95. The silicon-rich product with a particle size of less than 1 μm has a silica content of 57.57% can be used as high-value micro-sphere silica. The research results revealed in this work offers a potential and environmentally industrial treatment technique for the BCFA.

**Keywords:** brown corundum fly ash; water leaching; particle size separation; alumina; silica; potassium sulfate

## 1. Introduction

Brown corundum, which is composed of 95–97% Al$_2$O$_3$, and small amounts of Fe and Si, is widely used in refractories and grinding wheels because of its distinctive self-sharpness and strong grinding force [1]. Brown corundum is produced from high-quality bauxite and other raw materials in an electric arc furnace at a temperature higher than 2250 °C [2]. In this high-temperature process, dusts are exhausted with flow gas, and then the brown corundum fly ashes (BCFA) are collected by a dedusting technique [3,4]. At present, the theoretical annual outpour of BCFA is 400,000–500,000 tons [5], the BCFA concentration in untreated flow gas from a 7500 kVA brown corundum electric arc furnace was reported at 1500 mg/m$^3$ [6]. Hence, abundant of BCFA have been produced each year. BCFA has an average particle size of less than 1 μm and is mainly composed of K, S, Al and Si, which impedes its further utilization. Now, most of the BCFA wastes are stacking on the ground, which may result in potential water pollution by leaching or other environmental issues [3,4]. Therefore, utilizing and recovering BCFA is an urgent challenge for the development of the brown corundum industry.

The utilization of BCFA has attracted many researchers' interests. Li et al. [7,8] proposed a comprehensive utilization scheme for BCFA to obtain ultrafine aluminum hydroxide products in

theory. Zhou et al. [9] synthesized the zirconium mullite in an electric furnace with brown corundum fly ash. Quan [10] discussed the preparation of glass by using BCFA. Xue et al. [11] studied the casting performance of corundum–silicon nitride using BCFA as an additive agent. However, the impurities have a remarkable influence on the utilization of BCFA, especially potassium, sodium, etc. [12,13]. On the other hand, researchers have committed to recovering aluminum resources from various high aluminum solid wastes [14,15], such as extracting $Al_2O_3$ from coal fly ash, industrial aluminum dross [16,17], and aluminum packaging waste [18]. Furthermore, some high silicon raw materials are commonly used as raw materials in synthetic high-performance materials [19,20]. Due to the complex chemical compositions of BCFA, further attention should be paid to the recovery of the major elements, including K, S, Si, and Al for BCFA utilization.

In BCFA, the major phases are $K_2SO_4$, $Al_2O_3$ and $SiO_2$, in which the $K_2SO_4$ is soluble in water, while the alumina and silica can be separated by physical or chemical routes. So, environmental and non-polluting progress to recover the potassium sulfate, enrich and separate alumina and silica in BCFA, and the recovery ratio and effect factors were well-studied.

## 2. Materials and Methods

### 2.1. Materials

The studied BCFA was sampled from a typical brown corundum company in Guizhou Province, China. As shown in Table 1, the main chemical components of BCFA are $SiO_2$, $Al_2O_3$, and $K_2O$, and their contents in weight percentage (wt.%) are 40.12, 25.55, and 16.55 wt.%, respectively. The alumina-to-silica ratio (A/S) of the BCFA sample is 0.64.

**Table 1.** Contents of the main chemical components of brown corundum fly ashes (BCFA) (wt.%).

| $Al_2O_3$ | $SiO_2$ | $TiO_2$ | $K_2O$ | CaO | $Fe_2O_3$ | $Na_2O$ | $S_T$ | F | Cl | A/S |
|---|---|---|---|---|---|---|---|---|---|---|
| 25.55 | 40.12 | 0.92 | 16.55 | 0.32 | 4.88 | 1.21 | 3.85 | 0.51 | 0.36 | 0.64 |

The X-ray diffraction (XRD) pattern and distribution of the elements of BCFA by energy-dispersive X-ray spectroscopy (COXEM, ZTP10-08, Beijing, China) are shown in Figures 1 and 2, respectively. As demonstrated in Figure 1, the main phases of BCFA are amorphous silica, corundum, potassium sulfate, and quartz.

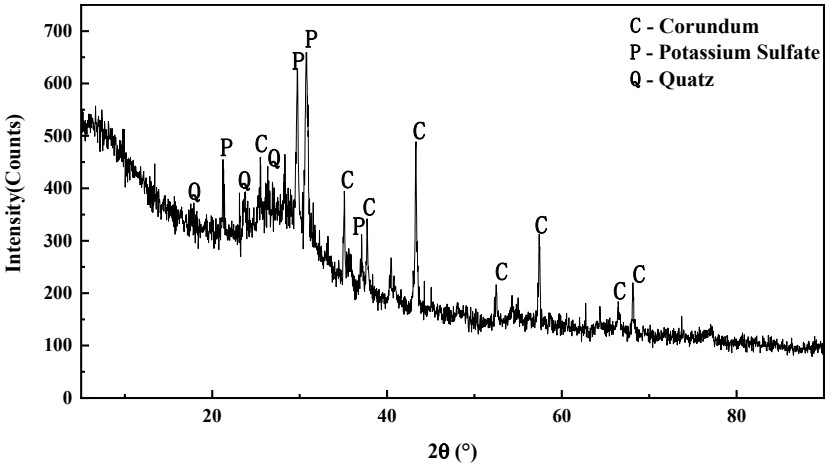

**Figure 1.** XRD pattern of BCFA.

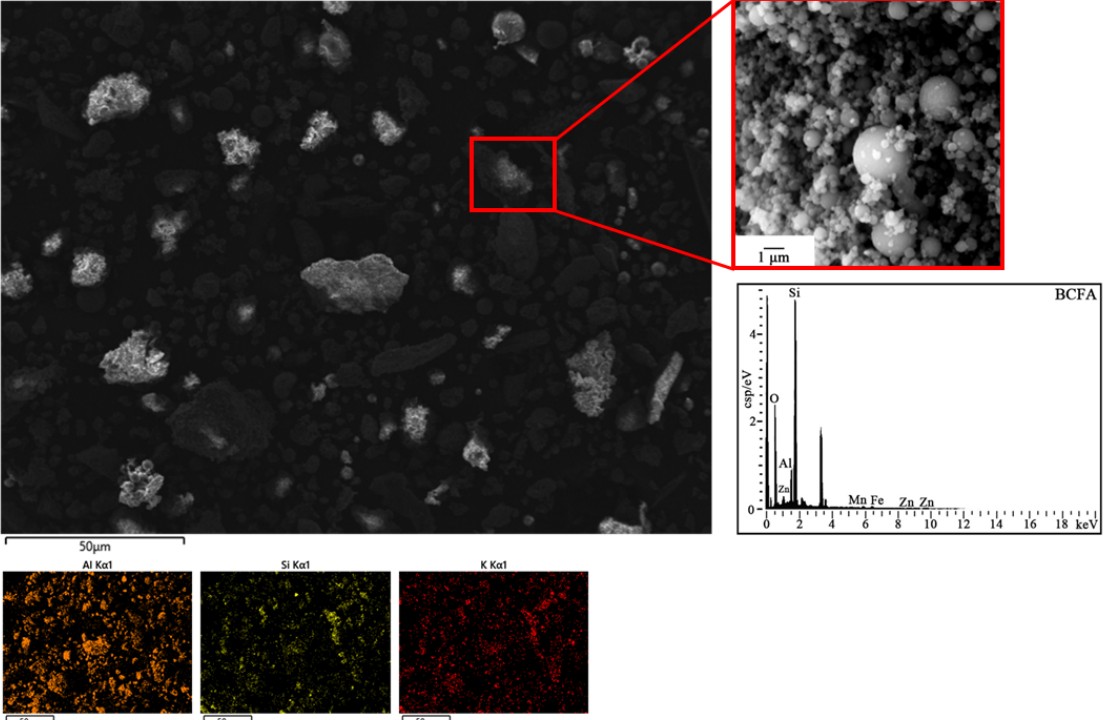

**Figure 2.** Energy-dispersive X-ray spectroscopy (EDS) profile of BCFA and the distribution of the elements Al, Si, and K.

From Figure 2, it can be seen that the major element of the particles with sizes larger than 1 μm is Al, and the EDS result of the region containing silicon and potassium shows that the major elements of the particles with sizes less than 1 μm are Si and K.

### 2.2. Experimental Progress

BCFA (10 g) was leached in deionized (DI) water, alkali (Na$_2$O concentration = 110 g/L), and acid (1 N HCl) for 30 min with a liquid-to-solid ratio of 10 at 95 °C, and the compositions of the insoluble residues were analyzed. The BCFA was wet screened by sieves of various sizes in the range of 140–2800 mesh (96–1 μm) with DI water. The residue was dried at 105 °C for 24 h, and then the compositions of the different-size residues were determined.

### 2.3. Analysis Method

The content of Al$_2$O$_3$, Fe$_2$O$_3$ and SiO$_2$ were measured by EDTA Complexometric Titration method, potassium dichromate method and silicon molybdenum blue colorimetric method accordingly. The determination of total sulfur content is measured by the gravimetric method. The content of potassium oxide and sodium oxide and recovered potassium sulfate product were analyzed by X-ray fluorescence spectrometer (XRF, EDX-LE, Shimadzu, Japan). The particle size distribution of BCFA was analyzed by using a laser particle size analyzer (Zetasizer Nano ZS 90, Malvern, UK). Phase characterization of flue dust was performed on an X-ray diffractometer (XRD, X PertPowder, Cu Kα, Almelo, The Netherlands) with a scan step of 10 (°)/min, in the 2θ range from 0° to 100°. The microstructure and elementals distribution were analyzed using scanning electron microscopy with an energy dispersive X-ray spectrometer (COXEM, ZTP10-08, Beijing, China).

## 3. Results

### 3.1. Basic Properties of BCFA

#### 3.1.1. Dissolution Characteristics of BCFA

The solubility of BCFA in the different solvents is shown in Table 2, revealing that only 17.50 wt.% of BCFA was dissolved in DI water, while 40.80 and 63.15 wt.% of BCFA was dissolved in acid and alkali, respectively.

**Table 2.** The solubility of BCFA in deionized (DI) water, acid, and alkali (wt.%).

| Leaching Method | Soluble | Insoluble |
|---|---|---|
| Water | 17.50 | 82.50 |
| Acid (1 N HCl) | 40.80 | 59.20 |
| Alkali ($Na_2O$ concentration = 110 g/L) | 63.15 | 36.85 |

The alumina and silica contents of the residues after leaching BCFA in DI water, acid, and alkali were analyzed, and the results are shown in Figure 3.

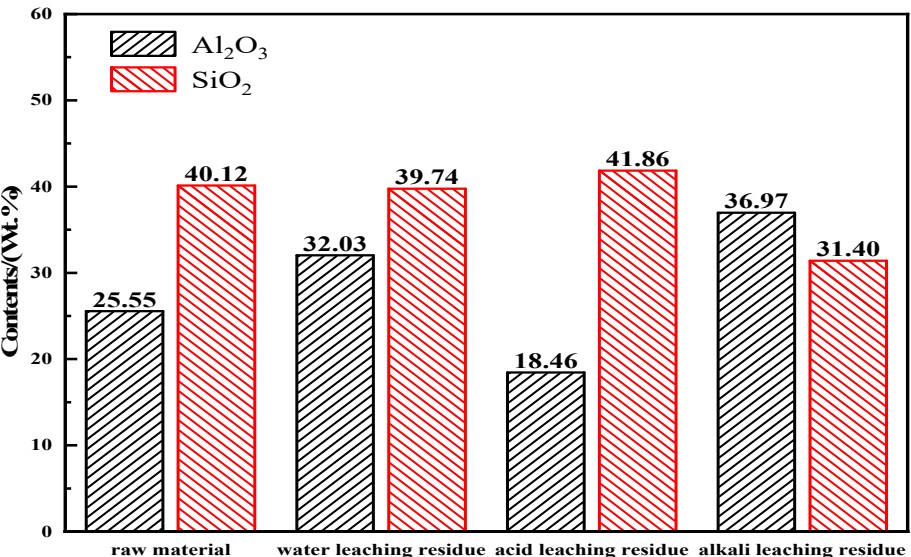

**Figure 3.** Compositions of water, acid, and alkali leaching residue.

It is known that corundum and quartz are insoluble in DI water at 95 °C, while potassium sulfate in BCFA is soluble in water, indicating that water leaching is a promising process to separate potassium sulfate. Similar to water, potassium sulfate is soluble in an acid solution, but we found that the $Al_2O_3$ content also decreased, indicating that some of $Al_2O_3$ simultaneously dissolved in the acid. This is because the alumina of BCFA reacts with acid according to Equation (1). On the contrary, silica was found to dissolve in an alkali solution with the concomitant dissolution of a small amount of the alumina; therefore, both silica and alumina can react with the alkali, and the reactions are based on Equations (2) and (3), respectively. Thus, both acid and alkali leaching are ineffective for separating silica and alumina or enriching BCFA, and water leaching can be employed to separate and recover potassium sulfate from BCFA.

$$Al_2O_3 + 6HCl = 2AlCl_3 + 3H_2O \tag{1}$$

$$SiO_2 + 2NaOH = Na_2SiO_3 + H_2O \tag{2}$$

$$Al_2O_3 + 2NaOH = 2NaAlO_2 + H_2O \tag{3}$$

### 3.1.2. Particle Size Separation of BCFA

The particle size distribution of BCFA is shown in Figure 4, revealing that most of the particle size is distributed in the range of 200–500 nm.

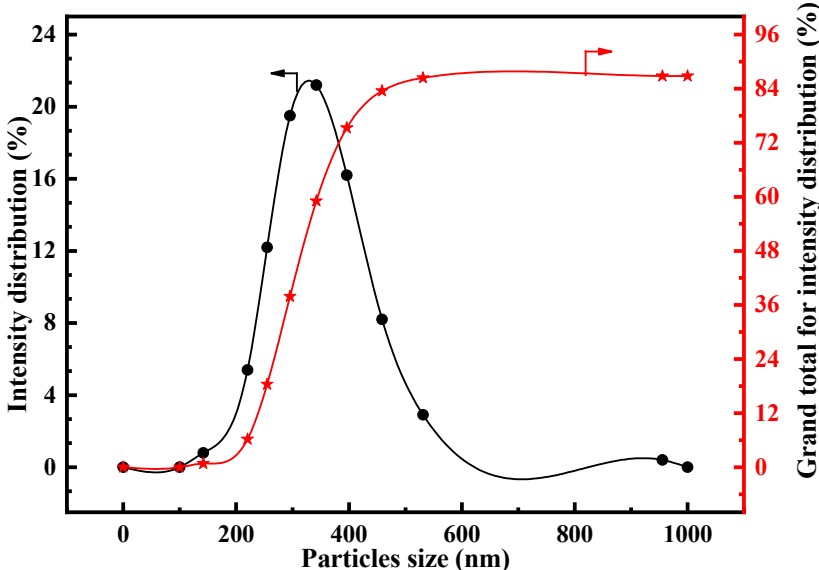

**Figure 4.** The particle size distribution of BCFA using a laser particle size analyzer.

The size distribution was determined by screening, and the results presented in Table 3 indicate that 85.12% of particles are smaller than 1 μm. Moreover, the main dimensions are 48–109 μm and 1–6.5 μm, which accounts for 4.11% and 5.30% of the particles, respectively.

**Table 3.** Grain size distributions of BCFA-Si and BCFA-Al.

| Mesh | BCFA-Si | | BCFA-Al | | | |
|---|---|---|---|---|---|---|
| | <1 μm | 1–6.5 μm | 6.5–28 μm | 28–48 μm | 48–109 μm | >109 μm |
| Percentage (%) | 85.12 | 5.30 | 2.67 | 0.55 | 4.11 | 2.25 |
| Grand total (%) | 85.12 | 90.42 | 93.09 | 93.64 | 97.75 | 100 |

According to the size distribution of BCFA, we used different sieve sizes to separate BCFA by wet screening, and then studied the alumina and silica contents of the samples with various size distributions. The results are shown in Table 4.

**Table 4.** Alumina and silica contents of BCFA after water leaching and wet screening with different sieve sizes (wt.%).

| Compositions | <1 μm | >1 μm | >6.5 μm | >13 μm | >18 μm | >28 μm | >48 μm | >109 μm |
|---|---|---|---|---|---|---|---|---|
| $Al_2O_3$ | 17.19 | 54.65 | 49.42 | 49.63 | 49.80 | 48.53 | 51.70 | 58.58 |
| $SiO_2$ | 57.57 | 11.04 | 12.40 | 13.60 | 13.99 | 14.70 | 13.26 | 15.62 |
| A/S | 0.30 | 4.95 | 3.99 | 3.65 | 3.56 | 3.30 | 3.87 | 3.75 |

It can be seen from Table 4 that when the size increases from 1 to 109 μm, the alumina content initially decreases from 54.65 wt.% and then increases to 58.58 wt.%, while the silica content continuously increases from 11.04 to 15.62 wt.%. The particles with sizes larger than 1 μm have the largest alumina-to-silica ratio (A/S) of 4.95. Moreover, it was found that the powder with particle sizes of less than 1 μm has the largest silica content of 57.57 wt.%. Therefore, the main component of the large

particles is alumina, while the small (<1 μm) particles mainly contain silica, indicating that a simple size separation method may achieve a silicon and aluminum enrichment in BCFA. Therefore, 1 μm size separation is an efficient method to separate the silica and alumina compounds in BCFA.

### 3.2. A Pollution-Free Progress to Recover $K_2SO_4$ and Separate Alumina/Silica in BCFA

Based on the above discussion, potassium sulfate can be separated by water leaching, which can be further recovered by vaporization, and the silica and alumina can be separated by the simple size separation method. Hence, a pollution-free process to recover potassium sulfate and separate alumina and silica in BCFA is proposed, which is summarized in Figure 5. As shown in Figure 5, the BCFA was leached in DI water, and then the solution was transferred for wet screening with a size of 1 μm, and the residue obtained after drying was named BCFA-Al. The filtrate containing the potassium sulfate solution and small-size BCFA was then subjected to a filtration process with a pore size of 0.2 μm. The residue collected after drying was BCFA-Si. The filtrate was then evaporated and the crystallized potassium sulfate product was produced.

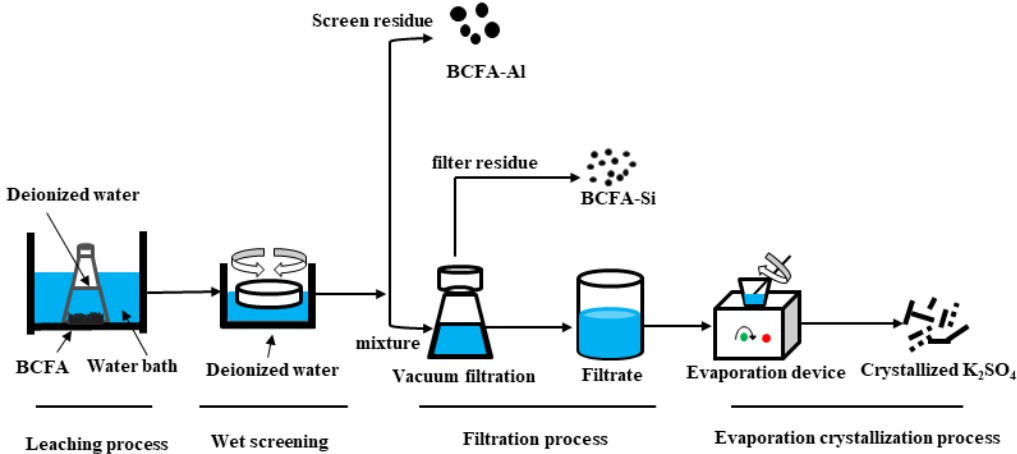

**Figure 5.** Schematic of the process for recovering $K_2SO_4$ and separating $Al_2O_3$ and $SiO_2$ from BCFA.

### 3.3. Separation of the BCFA-Al and BCFA-Si

The X-ray fluorescence (XRF) spectroscopy results of the collected BCFA-Al and BCFA-Si, which were obtained by leaching at 60 °C for 30 min with a liquid-to-solid ratio of 20 are listed in Table 5. As shown in Table 5, the alumina and silica contents in BCFA-Al are 54.65 wt.% and 11.04 wt.%, respectively, with an A/S of 4.95; raw materials with this composition can be considered for alumina production by the Bayer process or by sintering [21,22]. Moreover, some of the iron in BCFA-Al can be recovered by magnetic separation technology [23]. The alumina and silica contents in BCFA-Si are 17.19 wt.% and 57.57 wt.%, respectively; therefore, BCFA-Si can be regarded as a potentially high-valued microspheric silica raw material for widely used construction materials.

**Table 5.** Contents of the main compositions of BCFA-Al and BCFA-Si (wt.%).

|  | $Al_2O_3$ | $SiO_2$ | $Fe_2O_3$ | $TiO_2$ | CaO | MgO | $K_2O$ | $Na_2O$ | $SO_3$ | F | Cl |
|---|---|---|---|---|---|---|---|---|---|---|---|
| BCFA-Al | 54.65 | 11.04 | 16.21 | 3.95 | 2.58 | 0.57 | 2.04 | 0.57 | 0.72 | 0.83 | 0.25 |
| BCFA-Si | 17.19 | 57.57 | 2.56 | 0.36 | 0.35 | 0.81 | 3.05 | 0.53 | 0.19 | 0.04 | 0.00 |

The XRD patterns of BCFA-Al and BCFA-Si are shown in Figure 6. No obvious peaks due to potassium sulfate were observed in the XRD patterns of BCFA-Al and BCFA-Si, indicating that potassium sulfate was completely dissolved in the water. Moreover, a stronger amorphous peak was observed in the XRD pattern of BCFA-Si, and the intensities of the quartz diffraction peaks in the XRD

spectra of BCFA-Al are stronger, indicating that BCFA-Si was enriched with silicon, while BCFA-Al was enriched with aluminum.

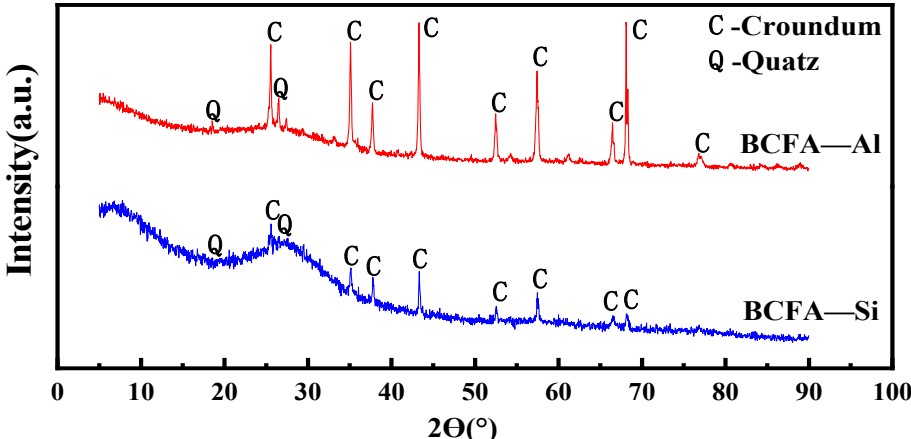

**Figure 6.** XRD analyses of BCFA after wet screening under 2800 meshes (BCFA-Al: wet screen residue, BCFA-Si: filter residue).

The scanning electron microscopy (SEM) and EDS results of BCFA, BCFA-Si, and BCFA-Al are shown in Figure 7. Figure 7a shows the SEM image of BCFA-Al, revealing that the average particle size of BCFA-Al is larger than that of BCFA. Moreover, the main element of BCFA-Al is Al. Figure 7b presents the SEM and EDS results of BCFA-Si. It can be seen that the BCFA-Si powders have an average size smaller than 1 μm and the main elements are Si and O. The results shown in Figure 7 indicates that the size separation method successively produced Al- and Si-enriched products, which is in agreement with the XRD patterns presented in Figure 6.

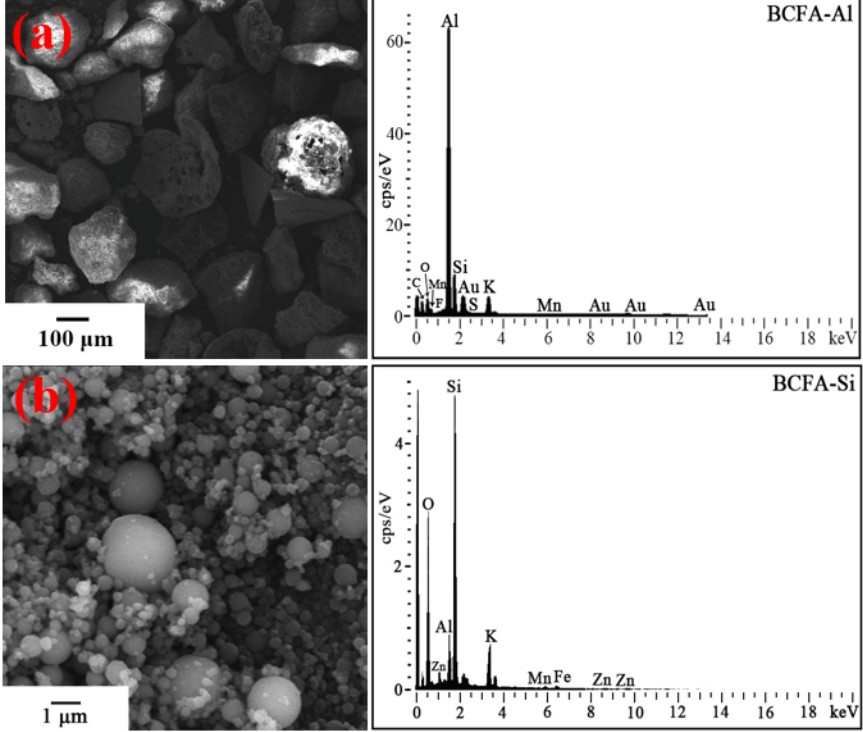

**Figure 7.** SEM and EDS of BCFA, BCFA-Al and BCFA-Si with wet screening BCDF under 2800 meshes (**a**) BCFA-Al: wet screen products, (**b**) BCFA-Si: vacuum filter residue.

## 3.4. Recovery of Potassium Sulfate from BCFA by Water Leaching

### 3.4.1. Effect of Temperature on Water Leaching BCFA

BCFA (10 g) was added to DI water in a water bath and stirred for various periods of time between 1 and 60 min at 30–90 °C. The XRD patterns of the crystallized products from leaching at 60 °C and 90 °C are shown in Figure 8. The phases of crystallized products are mainly potassium sulfate, and impurity phases were not observed. Thus, the more recovery efficiency of potassium sulfate is obtained when the recovery efficiency of the crystallized product is more enough.

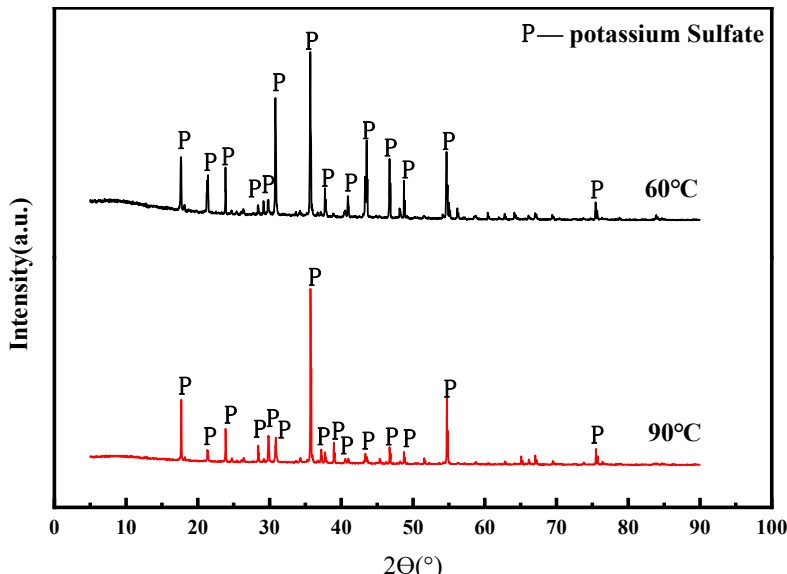

**Figure 8.** XRD spectra of the crystallized products of leaching at 60°C and 90°C (washing time 30 min, liquid-to-solid ratio 20).

The recovery efficiency of the crystallized product is calculated, which considers the change in mass before and after water leaching as follows:

$$\eta = \frac{m_O - m_L}{m_O} \times 100\%, \tag{4}$$

where $\eta$ is the recovery efficiency of the crystallized product, $m_L$ is the mass after water leaching, and $m_O$ is the mass of the raw material before water leaching.

From Figure 9, it can be seen that the recovery efficiency of the crystallized product first increases and then decreases with an increase in temperature, the value of the efficiency in the range 15–17% (Figure 9) suggest that only this fraction of the potassium sulfate/crystallized product was recovered. Theoretically, the recovery efficiency increases with increasing temperature, but impurities may affect the dissolution of potassium sulfate and finally affect the purity of the potassium sulfate product; e.g., the reaction of $Ca^{2+}$ with $SO_4^{2-}$ and the precipitation of undissolved $CaSO_4$. Therefore, water leaching should be conducted at 60 °C for a good recovery efficiency of potassium sulfate.

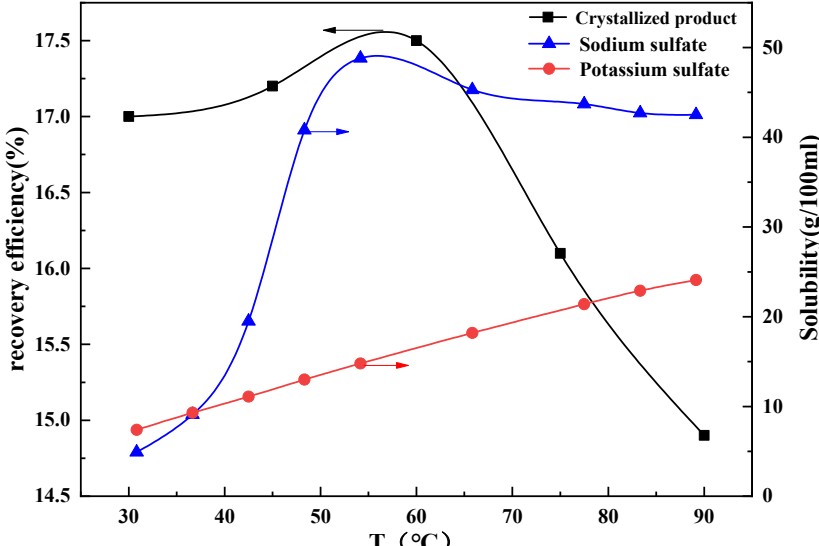

**Figure 9.** Effect of temperature on recovery efficiency of the crystallized product (leaching time 30 min, Liquid to solid ratio 20) and Solubility of $K_2SO_4$ and $Na_2SO_4$.

The chemical composition of BCFA shows that the leachate of the leaching process may contain soluble substances such as sodium and potassium salts ($Na_2SO_4$ and $K_2SO_4$). The theoretical solubility of potassium sulfate and $Na_2SO_4$ at various temperatures from 0 to 100 °C was calculated [24] by using FactSage 7.2. As shown in Figure 9, the solubility of $Na_2SO_4$ increases in the temperature range of room temperature to 40 °C and then decreases as the temperature continues to rise. For potassium sulfate, the solubility increases proportionally with increasing temperature. In conclusion, due to the decrease in $Na_2SO_4$ solubility, the recovery efficiency of the crystallized product also decreases.

### 3.4.2. Effect of Liquid to Solid Ratio and Time on Water Leaching BCFA

It can be seen from Figure 10a that the recovery efficiency of the crystallized products increases when the liquid–solid ratio increases from 5 to 20 $cm^3/g$ and with negligible changes when the liquid–solid ratio increases beyond 20:1. Therefore, the optimum liquid-to-solid ratio is 20:1 at 60 °C.

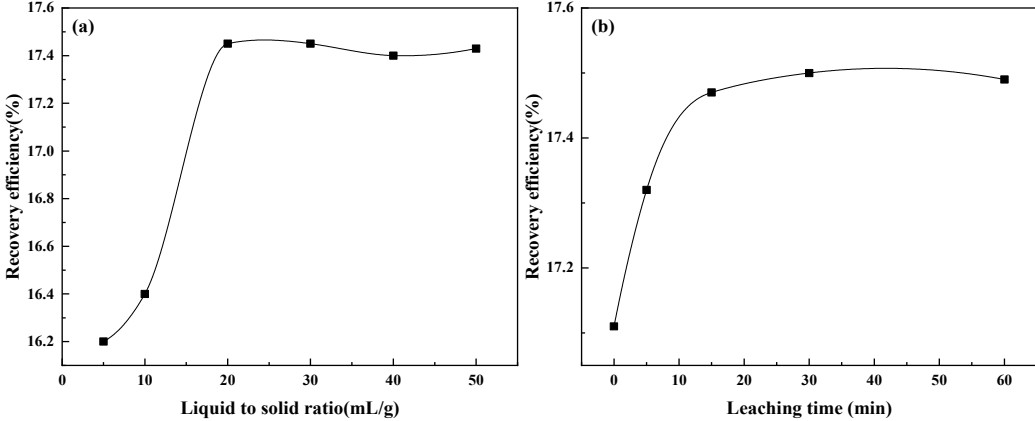

**Figure 10.** (**a**) Effect of liquid-to-solid ratio on recovery efficiency of the crystallized product (washing time 30 min, temperature 60 °C), (**b**) Effect of leaching time on recovery efficiency of the crystallized product (liquid-to-solid ratio 20, and temperature 60 °C).

Figure 10b shows the relationship between the recovery ratio and leaching time. The results show that the recovery efficiency of the crystallized product rapidly increases as the leaching time increases from 1 to 15 min. After a leaching time of 30 min, the recovery efficiency slowly increases and then

decreases with increasing leaching time. The highest recovery efficiency is 17.50% at 30 min. Hence, at 60 °C with a liquid-to-solid ratio of 20, the optimum leaching time is 30 min.

### 3.4.3. XRF, SEM and EDS Analysis of Crystallized Production

The SEM and EDS maps of the obtained crystalized products are shown in Figure 11a,b, indicating that the main product is potassium sulfate with a lamellar structure. Moreover, sodium atoms were present in the EDS maps, and the sodium content decreased when the temperature increased to 90 °C.

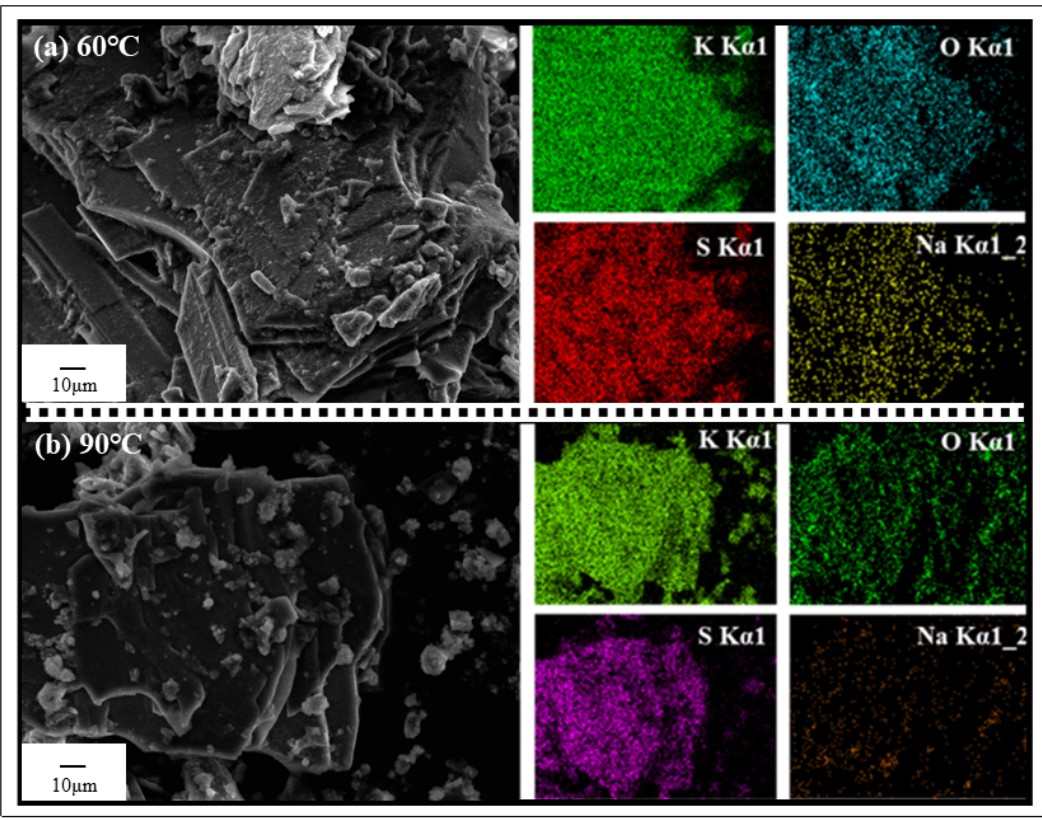

**Figure 11.** Electron microscopy and energy spectrum of crystal production after leaching at (**a**) 60 °C and (**b**) 90 °C.

The composition of the crystallized potassium sulfate product recovered at 60 °C after 30 min with a liquid-to-solid ratio of 20 is listed in Table 6. It can be seen that the main elements are K, S, and O, where potassium sulfate accounts for approximately 75.70% of the constituents. Therefore, 55.27% of potassium sulfate was recovered from BCFA.

**Table 6.** Contents of potassium sulfate product after evaporator crystallization (leaching time 30 min, leaching temperature 60 °C, liquid-to-solid ratio 20, wt.%).

| $K_2O$ | $SO_3$ | F | $P_2O_5$ | Cl | $Na_2O$ | CaO | $Al_2O_3$ | $SiO_2$ | MgO | $Fe_2O_3$ | MnO |
|---|---|---|---|---|---|---|---|---|---|---|---|
| 52.27 | 23.43 | 5.89 | 0.45 | 1.53 | 1.53 | 0.34 | 2.78 | 9.55 | 0.18 | 0.35 | 0.21 |

The potassium sulfate content of the product is relatively high, but fluorine and other impurities were detected in the potassium sulfate product, which required further purification. Zhan and Guo [25] proposed evaporating the purified filtrate step by step to obtain high-purity products. Moreover, alumina and silica impurities were detected in the recovered potassium sulfate, which may be due to the ultrafine size of BCFA. As a result, ultrafine silica and alumina passed through

the filter. A crystallized product that mainly contains potassium sulfate and a small amount of $Na_2SO_4$ can be separated by the improved methanol salting-out method [26].

## 4. Conclusions

By considering the leaching properties, as well as the size and composition distributions of BCFA, an environmentally friendly process was proposed to separate and recover the major compounds of potassium sulfate, $SiO_2$, and $Al_2O_3$ in BCFA. This simple recovery technique consisted of water leaching and size separation. The potassium sulfate content of the crystallized product was 75.70 wt.% after water leaching at 60 °C for 15 min with a liquid–solid ratio of 20:1. After the wet screening of the water-leached residue with a 1 μm sieve, the powder with particle sizes larger than 1 μm was found to contain alumina and silica contents of 54.65% and 11.04%, respectively, with an A/S of 4.95, which can be used for industrial alumina production. The remaining powder with particle sizes of less than 1 μm comprised 57.57 wt.% silica and 17.19 wt.% alumina, which can be considered for further material synthesis or as high-value microspheric silica raw materials for construction materials. This environmentally friendly and non-polluting process, which achieved the highest recovery of the major constituents of BCFA, may be up-scaled for industrial BCFA recovery.

**Author Contributions:** Conceptualization, Y.L.and Q.L.; methodology, Q.L. and Y.X.; software, L.W.; validation, Y.L., Q.L. and Y.X.; formal analysis, C.C.; investigation, C.C. and Y.X.; resources, J.L.; data curation, Y.X.; writing—original draft preparation, Y.X. and Q.L.; writing—review and editing, Y.X. and Y.L.; visualization, C.C.; supervision, J.L.; project administration, C.C.; funding acquisition, J.L. All authors have read and agreed to the published version of the manuscript.

**Funding:** This research was funded by National Natural Science Foundation of China, grant number U1812402, 51574095, 51664005, 51774102 and 52074096. Guizhou Alumina Production Technology and Technology Science and Technology Innovation Talent Team Project (Yankehe Talent Team Giant [2015] No. 4005, Yu Kehe Platform Talent [2017] No. 5788 and the Cooperation Talent Group of Guizhou Department [2017]5626). Guizhou Metallurgical Resources Comprehensive Utilization Engineering Research Center Project (Xu Jiaohe KY Ziju [2015]334), and Guizhou University Postgraduate Innovation Fund (Research Institute of Technology 2016018).

**Conflicts of Interest:** The authors declare no conflict of interest.

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
