# Peer review of "Recovery of K2SO4 and Separation of SiO2/Al2O3 from Brown Corundum Fly Ash"

_metals, doi:10.3390/met10121603_

Round 1

Reviewer 1 Report

This manuscript reports on the recovery of K2SO4 from brown corundum fly ass(BCFA) by simple leaching. The alumina and silica remained in the leaching residues were separated by size classification. After the following points are revised, the manuscript can be accepted for publication.

  1. Dissolution of alumina and silica should be explained on the basis of amphoteric and basic oxide nature, respectively.
  2. It is necessary to explain how the presence of Ca(II) affects the dissolution of K2SO4
  3. It is well known that the solubility of Na2SO4 decreases with increasing temperature.
  4. Therefore, it is necessary to check again the calculated results

Correct English errors as follows

  1. Title : Change the tile as Recovery of --- and separation of
  2. Line 54 : to recover
  3. Line 99 : promising
  4. Line 116 : which were obtained
  5. Line 126 : silica and alumina
  6. Line 149 : Figure 5
  7. Eq. (2) should be deleted.
  8. Line 157 : Ca2+
  9. Line 184 : shown-> show
  10. Line 185 : present
  11. Line 190 : recovered
  12. Line 224 : which were obtained

Author Response

 Response to the Reviewer’s Comments

We would like to thank you for your helpful comments. We have followed all the suggestions and made revisions accordingly. The list below gives the details of the changes that we have made in the revised manuscript.

Comment 1: Dissolution of alumina and silica should be explained on the basis of amphoteric and basic oxide nature, respectively.

Respond: According to your suggestion, the dissolution of alumina and silica have been explained on the basis of amphoteric and basic oxide nature, and relevant description has been listed in revision. More detail can be found in 3.1 section in revised manuscript.

Comment 2: It is necessary to explain how the presence of Ca(II) affects the dissolution of K2SO4.

Respond: Thanks for your helpful comment. The Ca(II) will affect the purity of K2SO4, where Ca2+ may react with react with SO42- and produce the slightly soluble CaSO4, and the undissolved CaSO4 precipitate will further reduce the purity of potassium sulfate product. The explanation has been added into the revision.

Comment 3: It is well known that the solubility of Na2SO4 decreases with increasing temperature. Therefore, it is necessary to check again the calculated results.

Respond: There is no doubt that the solubility of sodium sulfate decreases with increasing temperature, but the theoretical calculation is shown the solubility of sodium sulfate increases with the temperature increasing when the temperature range is 0℃ to 50℃, then it decreases with the temperature increasing. The calculation progress of solubility is as follows:

The theoretical solubility of sodium sulfate and calculated result is shown in table1:

Table1 The theoretical solubility of sodium sulfate and calculated result

Temperature (℃)

[1]/ (g/100 ml)

S

0

4.9

4.86

10

9.1

9.08

20

19.5

19.38

30

40.8

40.06

40

48.8

48.82

50

46.2

46.15

60

45.3

45.27

70

44.3

44.21

80

43.7

43.65

90

42.7

42.13

Where is the standard solubility of sodium sulfate, S is the result of theoretical calculation.

[1] Dalun Ye. Manual of Thermodynamic data for Practical Inorganic Materials[M]. Metallurgical Industry Press, 1981.

Comments 4:

Correct English errors as follows

  1. Title : Change the tile as Recovery of --- and separation of
  2. Line 54 : to recover
  3. Line 99 : promising
  4. Line 116 : which were obtained
  5. Line 126 : silica and alumina
  6. Line 149 : Figure 5
  7. Eq. (2) should be deleted.
  8. Line 157 : Ca2+
  9. Line 184 : shown-> show
  10. Line 185 : present
  11. Line 190 : recovered
  12. Line 224 : which were obtained

Respond: Thanks for your carefully suggestion, the mentioned mistakes have been corrected.

  1. Title: it has been changed as “Recover of K2SO4 and separation of SiO2/Al2O3 from brown corundum fly ash”
  2. Line 54: it has been modified.
  3. Line 107: it has been modified.
  4. Line128-130: it has been modified.
  5. Line137: it has been modified.
  6. It has been deleted.
  7. It has been deleted.
  8. Line232: it has been modified.
  9. Line203: it has been modified.
  10. Line233: it has been modified.
  11. Line238: it has been modified.
  12. Line155: it has been modified.

Reviewer 2 Report

Manuscipt metals-996148 is devoted to research on potassium recovery and separation of silica and alumina fractions originating from brown corundum fly ash (BCFA). The authors investigated the method in which BCFA is firstly leached in water, then wet screened and filtered. Potassium sulfate product is crystallised from the remaining solution.

There are some important issues that need to be clarified during the review process. Therefore, I cannot recommend to publish the manuscript in the current form.

Firstly, quality of English needs to be improved. There are many sentences which meaning is unclear, eg.
- line 102 “So, no significant separation and enrichment of Si and Al after the acid or alkali leaching”
- line 115 “we studied the alumina and silica contents of different size of samples which obtained by wet screening with different size of sieves”

Deeper analysis of the obtained results should be done. It is not clear how the components of BCFA are shared between respective fractions, i.e. Al-enriched fraction, Si-enriched fraction and crystalline product. Influence of investigated parameters on recoveries should be done. Also composition of the solutions obtained after leaching should be provided

- line 146 equation (1) – it is not recovery efficiency of potassium sulfate but mass loss during leaching. It would be K recovery if mO the mass of K2SO4 in raw material and mL mass of K2SO4 in water leached solid

- line 110 – Figure 3 is missing

- line 170-172 – the value of liquid to solid ratio should be provided either as a ratio (eg.10:1) or with units (cm3/g]

- line 194 – Table 5 – time unit is missing

- line 215 – 1 or 11 µm?

- line 232 – Table 6 – unit of content is missing

- line 246 – um --> µm

Author Response

Response to the Reviewer’s Comments

We would like to thank you for your helpful comments. We have followed all the suggestions and made revisions accordingly. The list below gives the details of the changes that we have made in the revised manuscript.

Comment 1: Firstly, quality of English needs to be improved. There are many sentences which meaning is unclear, eg.

- line 102 “So, no significant separation and enrichment of Si and Al after the acid or alkali leaching”

- line 115 “we studied the alumina and silica contents of different size of samples which obtained by wet screening with different size of sieves”

Respond: Thanks to you for your helpful comments. The sentences which meanings are unclear have been modified, please see section 3.1 of manuscript. (Line102 has been modified as line 112-113, line 115 has been modified as line 128-130).

Comment 2: Deeper analysis of the obtained results should be done. It is not clear how the components of BCFA are shared between respective fractions, i.e. Al-enriched fraction, Si-enriched fraction and crystalline product. Influence of investigated parameters on recoveries should be done. Also, composition of the solutions obtained after leaching should be provided.

Respond: In this work, the technique for separating K2SO4, as well as enriching the silica and alumina minerals was supposed after carefully investigating of the minerals, and chemical nature of the BCFA, where we found most of the K2SO4 can be dissolved into the water, and the silica and alumina contents in the BCFA with different size distribution are discrepant. Therefore, we proposed the method for recovering or enriching the K, S, Si and Al element in BCFA, which is given in Figure 5.

The influence of investigated parameters on recoveries has been done in work, which listed in 3.4.1 and 3.4.2 sections.

In this work, we used the mass loss of BCFA after leaching as the K2SO4 recover ratio, because the K2SO4 is the main water solubility in BCFA, which was identified by the XRD of BCFA. Moreover, the XRD pattern of the crystallized product only exhibit the diffraction peaks of K2SO4, which further proved that the mass loss of during the water leaching process can be used as the standard to represent the K2SO4 recover ratio. 

Now we are still working the deeper investigation of the utilization of BCFA, e.g. improve the purity of the crystallized product, etc., which may be carried out soon in our next work.

1- line 146 equation (1) – it is not recovery efficiency of potassium sulfate but mass loss during leaching. It would be K recovery if mO the mass of K2SO4 in raw material and mL mass of K2SO4 in water leached solid

2- line 110 – Figure 3 is missing

3- line 170-172 – the value of liquid to solid ratio should be provided either as a ratio (eg.10:1) or with units (cm3/g]

4- line 194 – Table 5 – time unit is missing

5- line 215 – 1 or 11 µm?

6- line 232 – Table 6 – unit of content is missing

7- line 246 – um --> µm

Respond:

  1. The equation(1) (now is equation 4) is surely the mass loss during leaching, and we used it to stand the recovery of potassium sulfate because the XRD result of the crystallized product shows that the potassium sulfate is the main composition of the crystallized product.

The mentioned mistakes have been corrected.

2.Line121: it has been added as Figure 4.

3.Line218-220 : it has been modified.

4.Line 242:it has been modified.

5.Line 259: it has been modified.

6.Line 163: it has been modified.

7.Line 259: it has been modified.

Reviewer 3 Report

Brown corundum fly ash (BCFA) is a unwanted by-product of brown corundum production, an industrially important good. Annually, about 500000 tons of BCFA are outpoured with concentrations of up to 1.5g/m3 and particle sizes <1µm. It is highly desirable to recover BCFA properly and develop utilisation paths for environmental reasons.

After a convincing introduction, however, the manuscript becomes more difficult to read. Some little linguistic issues need some spell checking, but that's not really the point here, it's not the English that makes the difficulties.

The sequence in the Results section (3.) may be a reason for the difficulties. The authors propose a complete recovery route for the BCFA, and they should place this at the center of their exposition, i.e., start the section with the general description of the process (now 3.3), and decorticate the process in chronological (or other consistent) order afterwards. Surely the K2SO4 extraction is at the center of their interest, but I would, intuitively, place the chapters concerning K2SO4, its leaching and its characterisation after the characterisation of the silicon- and aluminium containing partitions.

Besides, the numbering of the sections is wrong, but that's easily corrected (... 3.4 - 3.4.3 - 3.3.3 - 3.5 - 3.4).

Only in the conclusion it becomes clear why the order had been chosen as it is in the Results section: The process has not been presented right at the beginning (as I would suggest), as the proposed process is the result of preliminary studies on the BCFA composition and the K2SO4 leaching process. While a paper does not need to follow the chronological order of investigation -- as chosen by the authors -- it may be a valid choice. 

I would like the authors to consider another sequence, starting with the presentation of the whole procedure proposition and presenting the arguments for it later. They may stick to the current sequence, but should make the aim of the preliminary investigations clearer right in the beginning.

I would consider this paper for publication in this or a more adequate mining or chemical processing journal after minor revision. The results are not particularly exciting, astonishing or original, it is solid, down-to-earth waste processing, based on long-established methods and analytical tools, but it's nevertheless to be made known among the concerned communities of mining and mineral processing scientists and engineers.

Author Response

Response to the Reviewer’s Comments

We would like to thank you for your helpful comments. We have followed all the suggestions and made revisions accordingly. The list below gives the details of the changes that we have made in the revised manuscript.

Comment:

Brown corundum fly ash (BCFA) is a unwanted by-product of brown corundum production, an industrially important good. Annually, about 500000 tons of BCFA are outpoured with concentrations of up to 1.5g/m3 and particle sizes <1µm. It is highly desirable to recover BCFA properly and develop utilisation paths for environmental reasons.

After a convincing introduction, however, the manuscript becomes more difficult to read. Some little linguistic issues need some spell checking, but that's not really the point here, it's not the English that makes the difficulties.

The sequence in the Results section (3.) may be a reason for the difficulties. The authors propose a complete recovery route for the BCFA, and they should place this at the center of their exposition, i.e., start the section with the general description of the process (now 3.3), and decorticate the process in chronological (or other consistent) order afterwards. Surely the K2SO4 extraction is at the center of their interest, but I would, intuitively, place the chapters concerning K2SO4, its leaching and its characterisation after the characterisation of the silicon- and aluminium containing partitions.

Besides, the numbering of the sections is wrong, but that's easily corrected (... 3.4 - 3.4.3 - 3.3.3 - 3.5 - 3.4).

Only in the conclusion it becomes clear why the order had been chosen as it is in the Results section: The process has not been presented right at the beginning (as I would suggest), as the proposed process is the result of preliminary studies on the BCFA composition and the K2SO4 leaching process. While a paper does not need to follow the chronological order of investigation -- as chosen by the authors -- it may be a valid choice.

I would like the authors to consider another sequence, starting with the presentation of the whole procedure proposition and presenting the arguments for it later. They may stick to the current sequence, but should make the aim of the preliminary investigations clearer right in the beginning.

I would consider this paper for publication in this or a more adequate mining or chemical processing journal after minor revision. The results are not particularly exciting, astonishing or original, it is solid, down-to-earth waste processing, based on long-established methods and analytical tools, but it's nevertheless to be made known among the concerned communities of mining and mineral processing scientists and engineers.

Respond:

We appreciate for your thoughtful review of our manuscript, and we believe your suggestion and the modifications based on them will significantly stronger this manuscript.

We modified description and the title of most sections, and which may elevate the readability of this manuscript. In this work, the technique for separating K2SO4, as well as enriching the silica and alumina minerals was supposed after carefully investigating of the minerals, and chemical nature of the BCFA, where we found most of the K2SO4 can be dissolved into the water, and the silica and alumina contents in the BCFA with different size distribution are discrepant. Therefore, we proposed the method for recovering or enriching the K, S, Si and Al element in BCFA, which is given in Figure 5.

Reviewer 4 Report

[1] The paper is well written but I have the problem of understanding Figure 4 - - please explain.

[2] In Figure 5, I see for the first time sodium sulfate! Where it comes from? Please explain.

[3] On line 165, I see "Factsage 7.2" which I do not understand. Please explain.

Author Response

Response to the Reviewer’s Comments

We would like to explain the question for yours. The list below gives explain of the your questions.

Comment 1: The paper is well written but I have the problem of understanding Figure 4 - - please explain.

Respond: Figure 4 is the schematic of the BCFA recover and utilization process, which aimed for recovering potassium sulfate, as well as separating and enriching the Al2O3 and SiO2 minerals in BCFA.

Firstly, mixed the BCFA with DI water with a certain ratio of liquid to solid, then the mixture was heated in a water bath at a certain temperature for a certain time, where most of the K2SO4 in BCFA will be dissolved into water. Secondly, the mixture was screened by a sieve with the pore size of 1 μm, the residue on the sieve was then named as BCFA-Al, which contains more Al compounds. Thirdly, the obtained filtrate was filtrated by a vacuum filter with the pore size of 0.2μm, which has a high Si content and is named as BCFA-Si. At last, the filtrate after vacuum filtering was evaporated and produced the crystallized potassium sulfate.

Comment 2: In Figure 5, I see for the first time sodium sulfate! Where it comes from? Please explain.

Respond: A little amount of sodium is presented in BCFA, which is proved by XRF result. We discussed the sodium sulfate in Figure 5 because it may affect the purity of K2SO4.

Comment 3: On line 165, I see "Factsage 7.2" which I do not understand. Please explain.

Respond: Factsage 7.2, which is a powerful software ( 7.2 means the version of the Software) for thermodynamic calculating. We employed it to calculate the theoretical solubility of sodium sulfate and potassium sulfate, including the standard Gibbs generated energy, and the calculation of theoretical solubility as follows:

The theoretical solubility and calculated result of sodium sulfate is shown in table1 and table 2:

Table 1 The theoretical solubility and calculated result of sodium sulfate

Temperature (℃)

[1]/ (g/100 ml)

S

0

4.9

4.86

10

9.1

9.08

20

19.5

19.48

30

40.8

40.78

40

48.8

48.82

50

46.2

46.15

60

45.3

45.27

70

44.3

44.26

80

43.7

43.65

90

42.7

42.68

Table 2 The theoretical solubility and calculated result of potassium sulfate

Temperature (℃)

[1]/ (g/100 ml)

S

0

7.4

7.35

10

9.3

9.28

20

11.1

11.06

30

13.0

12.95

40

14.8

14.76

50

--

16.42

60

18.2

18.17

70

--

19.85

80

21.4

21.36

90

22.9

22.86

Where is the standard solubility, S is the result of theoretical calculation.

[1] Dalun Ye. Manual of Thermodynamic data for Practical Inorganic Materials[M]. Metallurgical Industry Press, 1981.

Round 2

Reviewer 2 Report

Some questions addressed in the first review report were addressed in the revised manuscript. But some of them still need to be improved.

Firstly, English still needs to improved. Sentences indicated as examles are improved, however there are still many other ones which have unclear meaning, e.g.:

Line 50-51 Based on the BCFA compositions, further attentions should be paid on the recovering of the major elements include K, S, 50 Si, and Al for BCFA utilization

Therefore, I would recommend extensive editing of language and style, e.g. using English editing service.

Explanation of equation (4) (no.1 in the first version) is still not clear. The value of the efficiency in the range 15-17% (Figure 9) suggest that only this fraction of the potassium sulfate/crystallized product was recovered. Other nomenclature should be used to the expression defined in eq.4.
What is the concentration of potassium in the solid obtained after water leaching?

Minor remarks:

Lines 19, 21, 259, 261 – um or µm?

Lines 147, 178 – 1 or 11?

Author Response

Response to the Reviewer’s Comments

I am very grateful to your comments for the manuscript. According to your advice, we amended the relevant part in manuscript. Some questions were answered below point by point:

Comment 1: Firstly, English still needs to improved. Sentences indicated as examles are improved, however there are still many other ones which have unclear meaning, e.g.:

Line 50-51 Based on the BCFA compositions, further attentions should be paid on the recovering of the major elements include K, S, 50 Si, and Al for BCFA utilization

Therefore, I would recommend extensive editing of language and style, e.g. using English editing service.

Respond: Thanks for your helpful comments. We have used the English editing service to modify description and the title of most sections, and which may elevate the readability of this manuscript.

Comment 2: Explanation of equation (4) (no.1 in the first version) is still not clear. The value of the efficiency in the range 15-17% (Figure 9) suggest that only this fraction of the potassium sulfate/crystallized product was recovered. Other nomenclature should be used to the expression defined in eq.4.

Respond: We have explained the equation (4) as follows:

Line 191-193:“The phases of crystallized products are mainly potassium sulfate, and impurity phases were not observed. Thus, the more recovery efficiency of potassium sulfate is obtained when the recovery efficiency of crystallized product is more enough.”

Line 204-206: “the value of the efficiency in the range 15-17% (Figure 9) suggest that only this fraction of the potassium sulfate/crystallized product was recovered.”

We have further explained the parameters appearing in equation (4), please see 3.4.1section, and the recovery efficiency of potassium sulfate has been recalculated as follows:

                       a=p1*m1/p0           (5)

where a is the recovery efficiency of potassium, m1 is the percentage of crystallized product mass, p0 and p1 are the contents of potassium oxide in raw BCFA and crystallized product, respectively.

The value of potassium sulfate recovery efficiency in the manuscript has been modified accordingly in line 246.

Comment 3: What is the concentration of potassium in the solid obtained after water leaching?

Respond: The potassium concentration in the solid after immersion is very low, and can be calculated from the data in Table 3 and 5. the concentration of potassium in the solid obtained after water leaching is 2.39 wt.%.

The calculation process as follows:

C=mL*(mBCFA-Si*C1+mBCFA-Al*C2)

Where the C, C1, and C2 are the concentration of potassium in the solid obtained after water leaching, the BCFA-Si, and the BCFA-Al, respectively, the mBCFA-Si and mBCFA-Al are the mass percentage of BCFA-Si and BCFA-Al in the solid obtained after water leaching, the mL is the mass after water leaching.
